# Application of Chitosan-Based Active Packaging with Rice Bran Extract in Combination with High Hydrostatic Pressure in the Preservation of Sliced Dry-Cured Iberian Ham

**DOI:** 10.3390/gels11070493

**Published:** 2025-06-25

**Authors:** Bruno Navajas-Preciado, Sara Martillanes, Javier Rocha-Pimienta, Jesús Javier García-Parra, Jonathan Delgado-Adámez

**Affiliations:** Centro de Investigaciones Científicas y Tecnológicas de Extremadura (CICYTEX), Instituto Tecnológico Agroalimentario de Extremadura (INTAEX), Avda. Adolfo Suárez s/n, 06007 Badajoz, Spain; javier.rocha@juntaex.es (J.R.-P.); jesusjavier.garcia@juntaex.es (J.J.G.-P.); jonathan.delgado@juntaex.es (J.D.-A.)

**Keywords:** non-thermal processing, eco-friendly packaging, chitosan gels, by-product, sustainable food

## Abstract

Iberian ham is a valuable product worldwide. At present, this product is mostly distributed and packaged in sliced form, which can result in loss of quality and safety. Moreover, non-biodegradable packaging exacerbates environmental problems. In this study, the application of active packaging based on a chitosan gel-like film and rice bran extract was investigated for the preservation of sliced Iberian ham. For this purpose, the packaging effectiveness on its own and in combination with high hydrostatic pressures was tested in comparison with untreated samples in refrigerated storage. The results showed that the active packaging used can maintain the reddish colour of sliced dry-cured Iberian ham, whereas browning took place in the control samples. Similarly, lipid oxidation of the product slowed, whereas protein oxidation was not affected by the packaging. This treatment also significantly reduces the number of microorganisms during storage.

## 1. Introduction

Iberian dry-cured ham is a valuable product. It is obtained from Iberian pigs, a breed reared outdoors in Mediterranean evergreen forests, mainly composed of *Quercus ilex* and *Quercus suber*, and which feeds mainly on acorns [1]. The manufacturing of Iberian pig products requires long rearing periods for pigs and long maturation times for hams [2]. These special manufacturing conditions mean that Iberian products are highly valued worldwide, with ham being the most valued.

Traditionally, whole pieces of Iberian ham are kept at room temperature until they are sold because their processing guarantees microbiological safety and stability [3]. However, shifts in consumer preferences towards convenience have led to an increasing demand for deboned and sliced formats [4]. This modern format, while more practical, presents challenges: slicing and packaging processes can lead to cross-contamination and enhance exposure to oxidative deterioration, ultimately reducing shelf life and product quality [5,6].

Oxidative degradation results in undesirable sensory attributes such as rancid odours and flavours, leading consumers to reject these products. While rancidity does not compromise safety, it significantly affects market acceptance and economic value. Moreover, extending shelf life aligns with broader global priorities, such as reducing food waste and enhancing food system sustainability [7]. Modified atmosphere packaging (MAP), vacuum packaging, and refrigeration are strategies commonly employed to limit microbial growth and oxidative reactions in sliced Iberian ham [8].

Recent advances have proposed the use of active packaging (AP) as a complementary or alternative strategy to MAP. AP systems interact with food and its environment to enhance quality and safety during storage. However, concerns remain regarding the potential migration of the active compounds from plastic matrices into food.

In previous research, olive leaf extracts were incorporated as natural antioxidants in active packaging but proved ineffective in preventing rancidity or microbial proliferation in ham [8], emphasising the need for more efficient compounds and delivery mechanisms.

Furthermore, conventional AP systems often rely on petroleum-derived plastics, which raises significant environmental concerns. These materials are typically single-use, non-biodegradable, and contribute to long-term environmental pollution [9]. Consequently, research is shifting toward biopolymers and biodegradable plastics derived from sustainable sources [10], including starch, collagen, cellulose, polyhydroxyalkanoates, and their derivatives [11].

Among these, chitosan, a biodegradable and non-toxic polymer derived from crustacean shells, has attracted significant interest. It exhibits antimicrobial properties and film-forming ability, and it is compatible with natural active agents [12]. Chitosan-based films and gels have shown promise for food preservation, especially when combined with plasticisers and cross-linkers such as citric or lactic acid. These organic acids form ionic and hydrogen bonds with chitosan, improving the film flexibility and structural integrity [13,14].

Ethanolic extraction of rice bran, a by-product of rice milling, allows the recovery of rice bran extract (RBE), a crude extract rich in natural bioactive compounds, particularly gamma-oryzanol groups. This approach not only valorises an agricultural residue but also yields an extract with demonstrated antioxidant, antimicrobial, and lipophilic characteristics [15] suitable for active packaging applications [16]. Its inclusion in active packaging has shown potential in preserving dry-cured meat products [5], although its efficacy is moderate compared to that of some essential oils. Nonetheless, its oil-based nature offers advantages in terms of compatibility with polymer matrices and ease of incorporation.

In addition to packaging innovations, non-thermal preservation technologies such as high-pressure processing (HPP) have demonstrated efficacy in microbial reduction without compromising product quality. HPP involves subjecting food to pressures between 100 MPa and 600 MPa for short durations. In Iberian ham, HPP has been shown to significantly reduce microbial populations and delay spoilage during storage [17]. The synergistic effect of high pressure and an active package based on chitosan and rice bran extracts has been previously studied for the reduction of *Listeria monocytogenes* [16].

The aim of this study was to assess the influence of eco-friendly active packaging made from chitosan and rice bran extract in conjunction with HPP on the microbiological and physicochemical characteristics of sliced dry-cured Iberian ham. This method aims to enhance safety and preserve quality during refrigerated storage, as well as extend the shelf life from three months to six months, without requiring cold storage.

## 2. Results and Discussion

### 2.1. Film Formation and HPP Effects

To evaluate whether HHP affects the inclusion and stability of RBE in chitosan formulations and potentially facilitates oil release via chitosan microfracture, we conducted a comparative FTIR spectral analysis focused on the key functional groups of both components. The inclusion of RBE within the chitosan hydrogel was first analysed, confirming that the incorporation was driven by the large molecular size of the oil particles retained inside the chitosan, with no evidence of chemical interaction with the chitosan matrix. Subsequently, the effect of HHP was evaluated to determine whether the applied treatment induced changes in the AP as a result of disruption of the physical structure of the chitosan network.

Characteristic RBE signals described by [18,19,20], primarily associated with aliphatic (C-H) stretching around the 2924 and 2853 cm^−1^ spectral zones, and a sharp ester carbonyl (C=O) absorption around 1740 cm^−1^ were evident in the untreated mixture (RBE–chitosan) and slightly intensified in the HHP-treated formulation. Similarly, the broad band near 3400 cm^−1^, typical of O-H and N-H stretches from chitosan, along with amide I (1650 cm^−1^) and amide II (1590 cm^−1^) bands [21], remained present post-treatment, suggesting that chitosan’s primary structural integrity was preserved. However, the modest decrease in RBE-specific transmittance in the HHP sample suggests the enhanced spectral visibility of the oil functionalities.

These observations strongly suggest that HHP treatment induces localised physical disruption within the chitosan matrix, leading to partial release or greater exposure of embedded RBE, without evidence of chemical degradation of the polymeric structure. This is supported by the decreased intensity of RBE-characteristic FTIR bands (2924–853, and 1740 cm^−1^) in the HHP-treated samples, while the spectral markers of chitosan (1650, 1590, and 3400 cm^−1^) remained unaltered, as shown in Figure 1. The preservation of these chitosan-specific bands confirmed the structural integrity of the hydrogel and excluded chemical modification as the mechanism underlying the observed changes. Instead, the enhanced visibility of RBE signals likely reflects pressure-induced reorganisation of the chitosan network, possibly through microfractures or increased matrix porosity, as shown in Figure 2, which facilitates infrared accessibility to the previously embedded oil. While this interpretation aligns consistently with the FTIR data, further confirmation using complementary material characterisation techniques would strengthen this mechanistic hypothesis.

### 2.2. Color Changes of Sliced Iberian Dry-Cured Iberian Ham Subjected to AP and HPP

Table 1 lists the colour changes of the sliced dry-cured Iberian ham. The parameter L*, which is related to the lightness of the product, increased significantly with respect to the control when AP and AP + HPP were applied (*p* < 0.05). Nevertheless, this initial variation disappeared after two months of storage (T1) and in subsequent analyses. At this time, the ANOVA revealed that there were no differences in lightness between the different treatments applied. On the other hand, in the control, the stability of the L* parameter was observed from T0 to T3, whereas at T4, this value increased significantly. Similarly, the AP samples showed higher L* values at T0 and T4 than the rest of the measurements. In the AP + HPP samples, time had no significant effect on the L* evolution.

The colour parameter a* is most closely related to the quality of meat, particularly dry-cured products, as it indicates the proportion of green to red in a sample. A bright red colour is associated with a high oxymyoglobin content, which is a positive determinant of meat quality [22], whereas greenish tones are associated with poor meat quality. All values of a* were positive for all treatments and study times, with the highest value obtained in the control at T0. The decreases produced by the treatments were not significant at any time point in this study. However, a significant decrease in the a* value was observed in both the control and in the AP + HPP treatments, particularly at T4. The P treatment maintained a constant a* value across all the sampling times. This stability suggests that AP treatment effectively inactivated oxidase enzymes and protected pigment molecules from oxidation, which is an essential factor for maintaining meat colour and, consequently, its marketability. As described in previous studies [23,24], pH, temperature, and pressure directly influence enzyme kinetics by inducing structural changes that affect the catalytic sites. These changes can significantly modulate enzyme activity, as demonstrated by [24] in their work on glucose oxidase produced by *Aspergillus niger*.

Parameter b* is related to the blue–yellow tones of the product. In this case, no treatment resulted in significant differences with respect to the control at any time point. However, both the control and active packaging evolved significantly. It should be noted that the RBE used in the production of the packaging has a yellowish hue, a colour that is generated, according to [25], by browning reactions that occur during extraction processes; the darker the bran pigments, the more aggressive is the extractant treatment. However, regardless of shade, it did not seem to affect the original colour of the sliced Iberian ham after being subjected to the packaging proposed in this study.

Chromaticity (C*) and hue angle (Hue) are variables that depend on the parameters a* and b*, respectively. Although there were no significant differences in C*, the hue angle showed significant differences between the treatments at T0. Similarly, an increase in hue over time was observed in both the control and AP treatment groups. An increase in the hue angle from values close to 40 to values above 60 indicates browning of the samples.

The colour of dry-cured ham is a multifactorial attribute influenced by subcutaneous fat coverage and pigmentation, the amount and distribution of intramuscular fat (marbling), and the degree of drying, all of which affect its consumer acceptability [26]. In this study, we obtained results similar to those obtained in our previous study [27], in which rice bran was directly added to conventional plastic packages.

### 2.3. Changes in Oxidation Status of Sliced Iberian Dry-Cured Ham with AP and HPP

To study the oxidative changes in sliced dry-cured Iberian ham, lipid oxidation and protein oxidation were measured as TBARS and based on the formation of carbonyl groups, respectively (Table 2). It can be observed that there was an initial decrease in TBARS values when both AP and AP + HPP were applied. This was due to the presence of antioxidants in the active packaging. Despite fluctuations in the measurements, owing to the variability of this particular product (the heterogeneous nature of Iberian ham owing to differences in fat infiltration and protein content [28]) and the reproducibility of the method, a protective effect was observed over time for the two treatments tested (AP and AP + HPP). Although the TBARS value increased significantly in the control, no such increase was observed in the treatments throughout the study period. The effects of the treatments on protein oxidation were difficult to estimate based on the data obtained. No significant differences were found between the control and treatment groups for up to 8 months of storage (T4). Moreover, the data were inconsistent throughout storage, although it seems that at T4 a pronounced increase in protein oxidation of sliced Iberian ham was observed.

The rice bran extract used in this study has been shown to increase the shelf life of different food matrices in previous studies. Thus, they can exhibit bioactivity and stability in oil-in-water emulsions [29] or prevent lipid oxidation owing to their specific structural characteristics [30]. When applied as part of an AP coating, the extract decreased lipid oxidation during long storage periods without modifying the sensory characteristics or volatile profile of the sliced dry-cured Iberian ham [5].

The effects of high pressure on lipid and protein oxidation in sliced Iberian ham were evaluated by [31], who found that HPP treatment significantly increased the TBARS values at T0. Moreover, the treated samples tended to show higher values than the control samples during refrigerated storage. The effect on protein oxidation was different from that on TBARS, and no differences were found that could be attributed to the application of HPP. A comparison of these data with those of the present study shows that the protective effects of chitosan and rice bran extract packaging extend even under high-pressure treatment.

### 2.4. Antimicrobial Effects of AP and HPP in Sliced Dry-Cured Iberian Ham

The microbiological changes in sliced Iberian dry-cured ham subjected to different treatments during storage are shown in Table 3. The presence of other pathogens and food spoilage organisms was also evaluated. Microbiological analyses showed an absence of *Salmonella* spp. and *L. monocytogenes* (in 25 g), and *C. perfringens*, *E. coli*, lactic acid bacteria, and *S. aureus* were below the detection limit for all storage times. The microbiological counts in the present study were within the accepted limits for human consumption established by [32].

AP with or without HHP treatment had an inhibitory effect (*p* < 0.05) on mesophilic bacterial counts. The applied treatment could induce sublethal changes; therefore, the effect was observed immediately after processing and during storage.

The psychrotrophic bacteria count was less than 1 log CFU g^−1^ in the T0, T1, T2, and T3 samples. However, the untreated samples showed an increase of more than 3 log CFU g^−1^ at T4, whereas AP continued to maintain counts below the limit of detection. The combination of AP and HPP reduced the counts compared to the control but was not able to keep the counts below the limit, as in the case of AP. One possible explanation for this difference is the known limitations of the mechanical stability of chitosan in humid environments. This issue has been reported in fields such as tissue engineering [33,34] and may affect AP*HPP treatment. High moisture can alter the chitosan structure, reducing the effectiveness of the biofilm in the controlled release of drugs (RBE). To address this problem, we followed the approach described by [34], incorporating plasticisers to reinforce the chitosan matrix and improve the stability and reproducibility of the film for controlled compound release applications (e.g., RBE). However, even after extended drying, residual moisture and the application of high pressure may still destabilise the film structure, leading to uncontrolled release and performance differences between the treatments. Another possible explanation (introduced previously in Figure 2) is the structure and flexibility of chitosan-based polymer gels which can undergo deformation under high-pressure processing. This is mainly due to gas absorption within the polymer matrix during compression, where gases act as plasticisers and may alter polymer crystallinity [35,36]. Rapid decompression and sudden gas expansion can lead to internal blistering and irreversible morphological changes. Figure 3 shows a clear variation in the functional groups on the tested chitosan (without RBE) before and after the same HHP treatment. Additionally, the combined effects of pressure, temperature, and time can alter the thermal, mechanical, and barrier properties of gels, thereby compromising their structural and functional stability [37]. This is a possible explanation for the phenomenon observed in our experiment between AP and AP*HPP treatments. Nevertheless, the fact that AP alone showed a stronger antimicrobial effect than combined AP*HPP treatment could be a promising finding, suggesting that industrial applications could benefit from using AP alone, which would simplify the process and reduce production costs.

According to the microbiological results, the initial mould and yeast counts were below the detection limits until T3 in all the samples (control, AP, and AP+ HPP). At T4, there was an increase in this group of microorganisms in the control group, which remained stable in the applied treatments. These results are in agreement with those of other studies, in which RBE demonstrated antifungal action [38]. Coliform counts were below the detection limit until T3. At T4, the counts in the control treatment were significantly higher than those in the treatment groups.

Taken together, these results highlight the importance of incorporating antimicrobial compounds into packaging materials to control specific microbial groups commonly associated with spoilage of high-quality dry-cured products, such as Iberian ham, synergising the activity of these molecules with other well-established preservation techniques in the food industry, such as modified atmospheric packaging, high-pressure processing (HPP), and cold storage.

### 2.5. Correlation Between Processing and Storage Parameters and Physicochemical and Biological Variables Studied

Finally, in order to analyse the influence of the type of packaging (control, AP, and AP+ HPP) and storage time (T0, T1, T2, T3, and T4) on the microbiological and physicochemical quality of sliced Iberian dry-cured ham, chromatic parameters (L*, a*, b*, C*, and Hue), oxidation status (TBA-RS and protein oxidation), and microbiological counts (mesophilic and psicrophilic aerobic bacteria, *E. coli* and coliforms, *C perfringens*, lactic acid bacteria, *S. aureus*, moulds and yeasts, *L. monocytogenes*, and *Salmonella* spp.) were determined. The results obtained (Table 4) from the two-way ANOVA for the type of packaging and storage time for each experimental parameter showed that the type of packaging had a significant influence (*p* < 0.05) on all the parameters investigated, except for the chromatic parameters, which means that the type of packaging does not modify the colour of the sliced Iberian ham. In principle, this is a positive factor because the appearance of slices is a fundamental parameter in the purchase of sliced Iberian ham [26]. In contrast, storage time was a significant factor (*p* < 0.05) for all parameters studied. Likewise, it was observed that the type of packaging × storage time interaction was also significant in protein oxidation and microbiological counts.

### 2.6. Limitations of the Study

This study has several limitations that should be acknowledged. First, sensory acceptability was not assessed, which limits the ability to correlate physicochemical parameters, such as colour and oxidative status, with consumer perception and purchase behaviour. This information is essential to validate the effectiveness of the proposed packaging strategies under real market conditions. Second, the potential migration of the active compounds into the food matrix was not evaluated. However, our research group previously addressed this issue in detail in recent studies [39,40], in which migration tests were performed according to the EU Food Safety Legislation (Regulation (EU) No. 10/2011) using food simulants, such as 95% ethanol (for fatty foods) and 50% ethanol (for oil-in-water emulsions) at 30 °C. These tests, using RBE with a group of γ-oryzanol as the active compound and its marker, demonstrated no detectable migration into the 50% ethanol simulant across various film formulations. This was attributed to the lipophilic nature of γ-oryzanol, the mild testing temperature, and the reduced extraction capacity of the water–ethanol mixture. In contrast, a significant release occurred only in the 95% ethanol simulant. Finally, this study focused on the design and preliminary assessment of an active packaging strategy as part of a broader development process. As such, it should be considered an intermediate step, providing the foundation for future studies aimed at validating these materials under real-world conditions, including scaling-up, sensory evaluation, and migration testing.

## 3. Conclusions

The present study illustrates that biodegradable active packaging (AP) composed of chitosan gel and rice bran extract (RBE), whether utilised independently or in conjunction with high hydrostatic pressure (HPP), effectively extends shelf life and maintains the quality of sliced Iberian dry-cured ham during prolonged refrigerated storage. The AP system contributed to colour stability by maintaining a* values and preventing the browning observed in control samples at eight months. Lipid oxidation was significantly reduced in all treated samples, with TBARS values consistently below 1 mg malondialdehyde (MDA)/kg, compared to 1.73 mg MDA/kg in the control at 8 months (T4). Protein oxidation exhibited a late-stage increase, particularly in the AP + HPP group, likely owing to pressure-induced alterations in the polymer matrix. The microbiological counts of spoilage organisms were significantly lower in AP and AP + HPP, with values remaining below the detection limits at T4, whereas the control demonstrated clear microbial growth. Notably, AP alone outperformed AP + HPP in certain parameters, suggesting that high pressure may compromise the matrix integrity and release dynamics. From a sustainability perspective, this packaging system facilitates the partial replacement of conventional plastic materials, aligning with circular economic objectives. Its multilayer format is compatible with existing vacuum and multilayer packaging workflows. However, integration into commercial processing lines necessitates further optimisation of the film-coating and sealing stages. Pilot-scale validation is required to assess the technical feasibility, reproducibility, and economic viability of applying chitosan gels in-line, particularly in high-throughput environments. Currently, the AP system is a promising solution for quality preservation and waste reduction in Iberian ham packaging. Future research should focus on scaling strategies, including process engineering studies to adapt the coating technology to industrial lines, in addition to sensory validation and real-matrix migration testing. Furthermore, considering the structural alterations observed in chitosan following high-pressure treatment, it is imperative to conduct further research to elucidate the modifications not only in functional groups (as indicated by FTIR) but also in essential physical parameters, such as crystallinity, mechanical performance (e.g., stress–strain behaviour), and viscoelastic properties. These factors are likely to pose significant challenges to industrial applicability and scalability.

## 4. Materials and Methods

### 4.1. Raw Material and Chemicals

Iberian pigs were reared outdoors in “montanera”, according to the Protected Denomination of Origin (PDO) [41] for “Iberian pigs that are destined for slaughter immediately after the exclusive use of acorns, grass, and other natural resources of the pasture, without the possibility of supplementary feeding”. Hams were obtained from the Montesano Extremadura Company (Badajoz, Spain). Four pieces of ham, each weighing approximately 7 kg, were used in this study. The ham slices were taken from the same areas of each piece and were randomly combined. Finally, the samples were divided into groups, with each group consisting of five samples. Slicing was manually performed using an expert slicer. The physicochemical characteristics of the sliced Iberian dry-cured ham studied were as follows: aw: 0.865 ± 0.007; pH: 5.65 ± 0.02; water: 36.1 ± 6.3%; protein: 30.1 ± 3.5%; and fat: 15.9 ± 3.8%. Chitosan from shrimp shells (C3646) and citric acid (W230618) were obtained from Sigma-Aldrich (Steinheim, Germany). Glycerol was purchased from Panreac (Castellar del Vallès, Spain). Rice bran extract (RBE) was obtained and characterised according to a previously optimised process [15].

### 4.2. Experimental Plan

The sliced dry-cured Iberian ham was homogeneously distributed into four groups, the first of which was a control sealed under vacuum with no AP. The second and third groups were packaged and coated with chitosan and rice bran extract films (AP), as described below, and sealed under vacuum. The third group combined films with HPP (AP + HPP). Finally, the fourth group was composed of HPP. This group was used in other studies [16], and because no complementary information was provided, we decided not to present the data in this study.

The samples were examined at the start of storage (T0), which refers to 24 h. As HPP treatments can cause sublethal damage to microbial populations under such conditions, a 24 h wait was allowed for cell death or cell viability to be re-established before assessment [42]. Additional assessments were conducted at two, four, six, and eight months (T1, T2, T3, and T4, respectively).

### 4.3. Film Formation and Packaging of Samples

Chitosan-based film-forming solutions were prepared as described [16], with 2% (*w*/*v*) chitosan dissolved in 2% (*w*/*v*) citric acid and 50% (*w*/*w*) glycerol (relative to chitosan) as plasticisers. Following sonication, a solution of 0.25 g RBE/100 g was added. Citric acid functioned as a mild crosslinker, enhancing the structural cohesion through ionic and ester interactions. Sixty grams of the solution was cast and allowed to dry in the dark at room temperature for 48 h. Although the crosslinking density was not quantified, the formulation facilitated a consistent film formation. After high-pressure processing, no macroscopic evidence of film disintegration or delamination was observed.

To simulate multilayer packaging, two chitosan-based biopolymers obtained under the previous conditions were used to coat 25 g of sliced ham on both sides and then sealed under vacuum in polyamide polyethylene packaging (20/100) provided by Plásticos Alvarez (Badajoz, Spain), with a thickness of 120 μm per bag. The control samples were similarly packaged without chitosan-based biopolymers.

#### Fourier Transform Infrared (FT-IR) Spectroscopy

FT-IR spectroscopy was employed to qualitatively monitor the incorporation of RBE into the chitosan polymer through the detection of characteristic functional groups exhibiting high-intensity absorption signals. Measurements were performed using a PerkinElmer instrument (Shelton, CT, USA). The UATR Two spectrometer was equipped with a single-beam ATR accessory and diamond crystal optics. Spectra were recorded in transmittance mode over the range of 4000–400 cm^−1^, with a resolution of 1 cm^−1^ and an accumulation of 16 scans per sample. Spectral data were processed and analysed using OriginPro 9.0 (OriginLab Software, Northampton, MA, USA).

### 4.4. High-Pressure Processing

The treatments were applied in a semi-industrial discontinuous hydrostatic unit (Hiperbaric Wave 6000/55, Hiperbaric, S.A., Burgos, Spain). Based on the information obtained in previous studies [43,44], the samples were pressurised at 600 MPa for 8 min using water at 10 °C as the pressure-transmitting medium. The times required to reach the maximum pressure of 600 MPa were 3 s and 50 s. Decompression of the vessel was instantaneous. The samples were then stored at 4 °C until further analysis.

### 4.5. Analysis of Dry-Cured Iberian Ham

#### 4.5.1. Chromatic Parameters

Lightness (CIE L*), redness (CIE a*), and yellowness (CIE b*) were evaluated using a Konica Minolta CM-5 spectrophotometer (Konica Minolta, Tokyo, Japan). Reflectance was measured using an illuminant D65 with a viewing angle of 10°. The aperture diameter was set to 30 mm. The hue angle (h° = tan (b*/a*)) and chroma (C*) (C = (a^2^ + b^2^)^0.5^) values were calculated using the above equations. Measurements were performed on five slices from each package and averaged [8].

#### 4.5.2. Oxidation Status

The oxidation status was estimated by the determination of thiobarbituric acid reactive substances (TBARS) and protein oxidation. They were measured using methods described in a previous study [27]. TBARS estimates lipid oxidation, and its values are expressed as mg malondialdehyde (MDA) kg^−1^ meat (mg MDA kg^−1^). Protein oxidation was assessed by estimating the carbonyl groups formed during incubation with 2,4-dinitrophenylhydrazine and was expressed as nmol carbonyls mg^−1^ protein.

#### 4.5.3. Microbiological Changes

Mesophilic and psychrophilic aerobic bacteria, *Escherichia coli*, coliforms, *Clostridium perfringens*, lactic acid bacteria (LAB), *Staphylococcus aureus*, moulds, and yeasts were measured as previously described [27]. However, according to the water activity measured in the ham, since the activity was just below the recommendation, it was decided to apply the following determinations: (1) The legislation of many countries requires listeria-free products. (2) It was also considered that the counts of mesophiles, psychrophiles, moulds, and yeasts determine the set of microorganisms included under aw requirements. (3) In the case of moulds and yeasts, the protocol used was that of microorganisms with aw < 95. The results were expressed as log colony-forming units (CFU) g^−1^. The detection limit was 1 log CFU g^−1^, with the exception of *S. aureus*, which had a detection limit of 2 log CFU g^−1^. The presence of pathogens such as *L. monocytogenes* and *Salmonella* spp. in 25 g of ham was also analysed at the beginning of storage. The results showed the absence of these pathogens; therefore, they were not analysed in the following days of storage.

### 4.6. Statistical Analysis

The means and standard deviations (SDs, *n* = 5) are reported. A multivariate analysis of variance (2-way ANOVA) was applied to data obtained using SPSS 21.0 (SPSS Inc., Chicago, IL, USA), taking into account the type of packaging (control, AP, or AP + HPP) effects, the storage time (T0, T1, T2, T3, and T4), and their interactions. One-way ANOVA was also applied to analyse the impact of the treatments applied and the development of the parameters during storage. When the ANOVA test was significant (*p* < 0.05), the means were compared using Tukey’s test.

### 4.7. Use of AI-Based Language Support Tool

ChatGPT was used (OpenAI, GPT-4o, 2024; San Francisco, CA, USA) to assist in partially generating Figure 2. The illustration was designed to represent the interaction between RBE and chitosan-based gel, and to depict how high-pressure treatment may lead to localized structural disruptions (microfractures) within the gel matrix, facilitating the release of RBE. The AI-generated figure was based on detailed chemical and mechanistic inputs provided by the authors. All content was thoroughly reviewed and edited for scientific accuracy, and the authors assume full responsibility for its final form.

## Figures and Tables

**Figure 1 gels-11-00493-f001:**
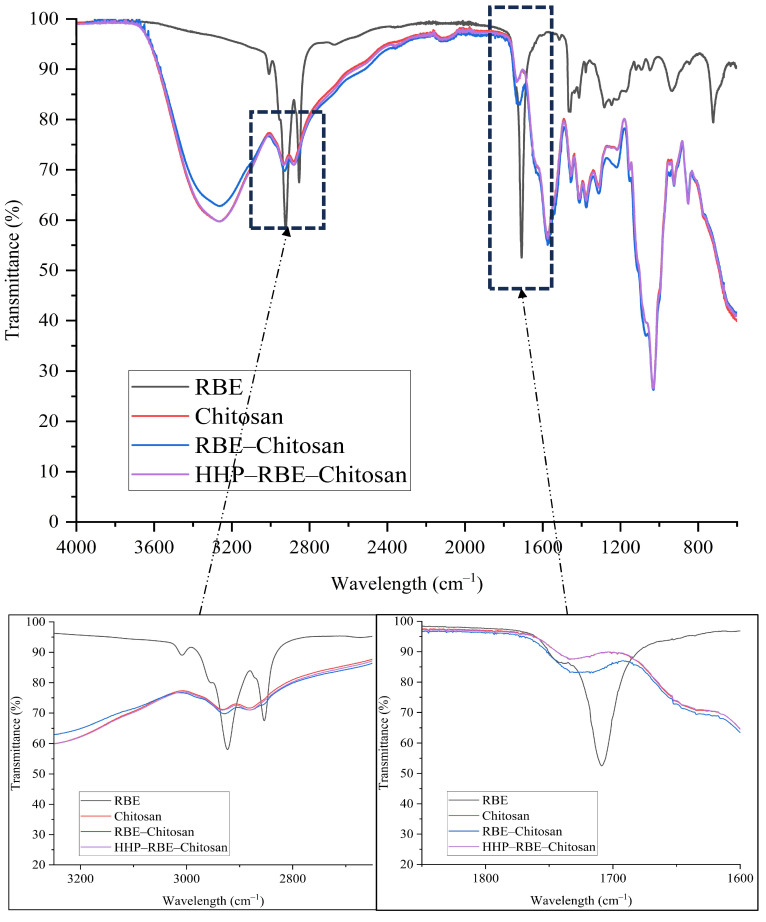
IR spectra were obtained for the raw materials (RBE and chitosan) and before and after HHP treatment. Amplified characteristic signals of the main functional groups of RBE.

**Figure 2 gels-11-00493-f002:**
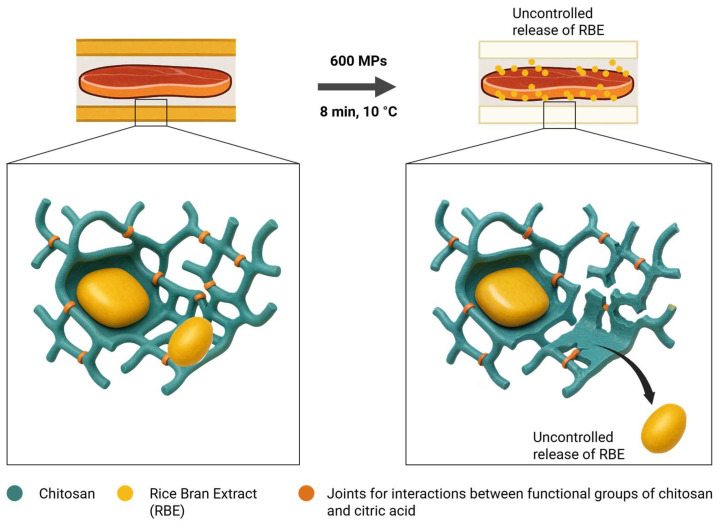
Effect of high-pressure treatment conditions on the stability of gel-like active packaging (chitosan, citric acid, and RBE).

**Figure 3 gels-11-00493-f003:**
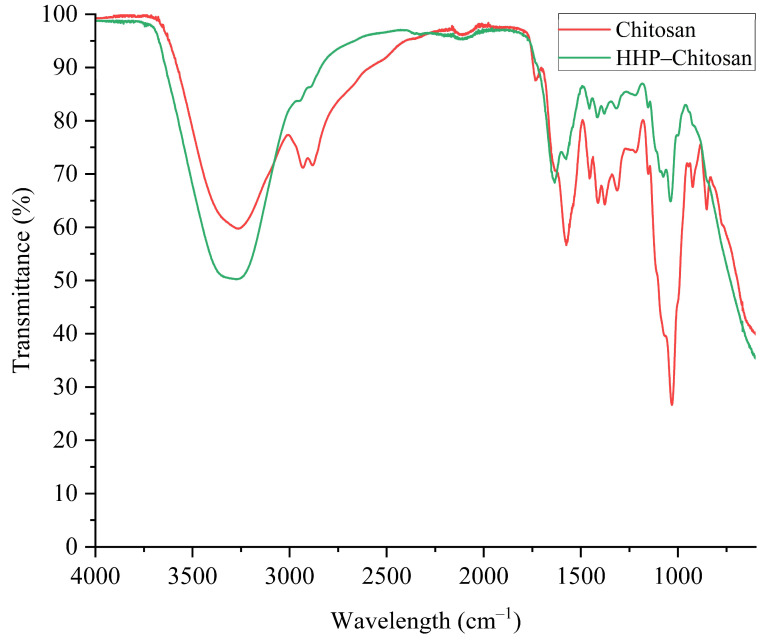
IR spectra obtained for treated and untreated chitosan with HHP and without RBE.

**Table 1 gels-11-00493-t001:** Colour changes in sliced dry-cured Iberian ham subjected to AP and HPP compared to the control.

**L***	**Control**	**AP**	**AP + HPP**	** *Significance* **
T0	47.7 ± 1.3 bB	53.8 ± 2.4 aA	51.1 ± 2.5 a	0.002
T1	48.8 ± 1.3 B	48.9 ± 2.3 BC	47.4 ± 2.0	0.391
T2	44.6 ± 1.2 B	45.7 ± 1.6 C	47.0 ± 2.5	0.154
T3	46.3 ± 1.5 B	46.8 ± 2.4 C	46.9 ± 2.8	0.902
T4	53.1 ± 5.2 A	52.6 ± 3.5 AB	50.7 ± 6.6	0.743
*Significance*	0.001	0.000	0.207	
**a***	**Control**	**AP**	**AP + HPP**	** *Significance* **
T0	8.0 ± 0.7 A	6.7 ± 0.9	7.2 ± 1.5 A	0.197
T1	6.1 ± 1.2 AB	5.8 ± 0.9	7.2 ± 1.0 A	0.106
T2	6.4 ± 1.1 AB	7.8 ± 0.9	7.1 ± 1.4 A	0.184
T3	7.7 ± 1.6 A	7.1 ± 1.7	7.1 ± 1.8 A	0.789
T4	4.4 ± 1.7 B	6.0 ± 2.3	4.7 ± 0.7 B	0.301
*Significance*	0.002	0.185	0.027	
**b***	**Control**	**AP**	**AP + HPP**	** *Significance* **
T0	6.9 ± 0.4 bAB	9.4 ± 0.9 aA	8.6 ± 2.4 ab	0.054
T1	6.8 ± 1.8 AB	6.8 ± 1.1 B	7.6 ± 1.1	0.586
T2	4.6 ± 1.4 B	6.2 ± 0.8 B	6.8 ± 2.4	0.124
T3	6.9 ± 1.6 AB	6.1 ± 0.9 B	6.4 ± 1.4	0.633
T4	8.0 ± 1.7 A	9.2 ± 2.7 A	7.9 ± 2.0	0.589
*Significance*	0.025	0.002	0.447	
**C***	**Control**	**AP**	**AP + HPP**	** *Significance* **
T0	10.7 ± 0.6	11.8 ± 0.8	11.4 ± 2.6	0.539
T1	9.2 ± 2.0	9.2 ± 1.0	10.6 ±1.3	0.234
T2	8.0 ± 1.5 b	10.1 ± 0.9 a	10.2 ± 1.4 a	0.032
T3	10.6 ± 1.7	9.5 ± 1.1	9.6 ± 2.1	0.544
T4	9.3 ± 1.6 a	11.2 ± 2.9	9.5 ± 1.6	0.318
*Significance*	0.066	0.067	0.497	
**Hue**	**Control**	**AP**	**AP + HPP**	** *Significance* **
T0	40.9 ± 2.1 bBC	53.8 ± 4.7 aA	48.7 ± 6.2 a	0.003
T1	47.2 ± 3.7 B	48.7 ± 4.8 AB	45.5 ± 3.8	0.487
T2	33.5 ± 7.4 C	38.5 ± 4.1 B	42.6 ±13.5	0.323
T3	41.2 ± 7.7 BC	41.0 ± 8.1 B	41.8 ± 4.5	0.981
T4	60.7 ± 11.5 A	56.59 ±11.0 A	56.4 ± 8.8	0.791
*Significance*	0.000	0.002	0.063	

a–b: Different letters in the same row indicate statistically significant differences (Tukey’s test, *p* < 0.05) between treatments. A–C shows significant differences (Tukey’s test, *p* < 0.05) among the storage times.

**Table 2 gels-11-00493-t002:** Oxidative status of sliced dry-cured Iberian ham subjected to AP and HPP compared to the control. Lipid oxidation was expressed as milligram malondialdehyde (MDA) kg^−1^ meat (mg MDA kg^−1^). Protein oxidation was expressed as nmol carbonyl mg^−1^ protein.

**Lipid Oxidation**	**Control**	**AP**	**AP + HPP**	** *Significance* **
T0	0.93 ± 0.20 aAB	0.70 ± 0.09 b	0.70 ± 0.02 b	0.025
T1	1.24 ± 0.58 AB	0.60 ± 0.14	0.70 ± 0.20	0.061
T2	0.67 ± 0.19 B	0.62 ± 0.02	0.81 ± 0.36	0.483
T3	1.12 ± 0.51 AB	0.60 ± 0.33	1.12 ± 0.36	0.105
T4	1.73 ± 0.77 A	0.89 ± 0.45	0.94 ± 0.40	0.062
*Significance*	0.036	0.440	0.233	
**Protein Oxidation**	**Control**	**AP**	**AP + HPP**	** *Significance* **
T0	5.6 ± 1.5 B	6.7 ± 2.7	9.3 ± 1.6 B	0.345
T1	9.3 ± 1.6 A	10.7 ± 1.8	11.5 ± 6.0 B	0.656
T2	7.8 ± 0.8 AB	6.2 ± 3.3	7.7 ± 4.9 B	0.749
T3	7.1 ± 2.4 AB	5.8 ± 2.8	6.4 ± 3.1 B	0.759
T4	6.0 ± 1.7 bB	12.4 ± 7.2 ab	27.0 ± 12.6 aA	0.011
*Significance*	0.016	0.046	0.001	

a–b: Different letters in the same row indicate statistically significant differences (Tukey’s test, *p* < 0.05) between treatments. A–B shows significant differences (Tukey’s test, *p* < 0.05) among storage times.

**Table 3 gels-11-00493-t003:** Microbiological counts of sliced dry-cured Iberian ham subjected to AP and HPP compared with the control. The results are expressed as log CFU g^−1^.

**Mesophiles**	**Control**	**AP**	**AP + HPP**	** *Significance* **
T0	1.56 ± 0.41 B	<1	<1	0.110
T1	1.62 ± 0.47 B	<1	<1	0.005
T2	<1	1.38 ± 0.76	<1	0.006
T3	1.09 ± 0.56 C	1.20 ± 0.75	<1	0.436
T4	3.57 ± 0.37 aA	1.05 ± 0.88 b	1.18 ± 0.67 b	0.000
*Significance*	0.000	0.173	0.056	
**Psicrophiles**	**Control**	**AP**	**AP + HPP**	** *Significance* **
T0	<1	<1	<1	-
T1	<1	<1	<1	-
T2	<1	<1	<1	-
T3	<1	<1	<1	-
T4	3.11 ± 1.70 a	<1	1.99 ± 1.71 b	0.016
*Significance*	0.000	-	0.002	
**Mould and Yeasts**	**Control**	**AP**	**AP + HPP**	** *Significance* **
T0	<1	<1	<1	-
T1	<1	<1	<1	-
T2	<1	<1	<1	-
T3	<1	<1	<1	-
T4	5.33 ± 0.25	<1	<1	0.000
*Significance*	0.000	-	-	
**Coliforms**	**Control**	**AP**	**AP + HPP**	** *Significance* **
T0	<1	<1	<1	-
T1	<1	<1	<1	-
T2	<1	<1	<1	-
T3	<1	<1	<1	-
T4	3.50 ± 0.23 a	1.95 ± 1.05 b	1.89 ± 1.01 b	0.018
*Significance*	0.000	0.000	0.000	

a–b: Different letters in the same row indicate statistically significant differences (Tukey’s test, *p* < 0.05) between treatments. A–C shows significant differences (Tukey’s test, *p* < 0.05) among the storage times.

**Table 4 gels-11-00493-t004:** Significance values for 2-way ANOVA.

	Packaged	Storage Time	Time Packaged
**L***	0.228	0.000	0.164
**a***	0.896	0.000	0.146
**b***	0.088	0.000	0.313
**C**	0.331	0.000	0.273
**Hue**	0.181	0.032	0.215
**Lipid oxidation**	0.000	0.010	0.130
**Protein oxidation**	0.004	0.000	0.000
**Mesophiles**	0.000	0.000	0.000
**Psicrophiles**	0.004	0.000	0.000
**Mould and yeasts**	0.000	0.000	0.000
**Coliforms**	0.005	0.000	0.000

## Data Availability

The raw data supporting the conclusions of this article will be made available by the authors on request.

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
