# Peer review of "Application of Chitosan-Based Active Packaging with Rice Bran Extract in Combination with High Hydrostatic Pressure in the Preservation of Sliced Dry-Cured Iberian Ham"

_gels, 2025, doi:10.3390/gels11070493_

Round 1
Reviewer 1 Report
Comments and Suggestions for Authors
The manuscript gels-3680771 presents an active material based on chitosan and rice bran extract (RBE), which was used for packaging a sliced ​​dry-cut Iberian ham. Additionally, the influence of high hydrostatic pressure (HPP) treatment on the durability of packaged food products was checked.
The described active material showed the ability to limit the development of undesirable microflora during the long-term storage of Iberian ham. The tested active packaging had no significant effect on the appearance (chromatic parameters) of the stored ham. The tested packaging and HHP treatment showed an ambiguous impact on the oxidative stability of fat and proteins. At various stages of storage, active packaging and HHP treatment limited the oxidative processes, in others it had no effect or even accelerated them. The description of these research results needs improvement. Much of the content presented in lines 200 to 228 refers to other studies, which should have been included in the introduction to the article. The description of the research results may refer to other studies, but not to such a large extent.
As the Authors rightly noted, the assessment of the usefulness of active packaging requires additional research in particular with regard to testing the migration and stability of active ingredients and their structures. The presented research results can be supplemented by a comparison of the antimicrobial activity of films with chitosan and rice bran extract before and after HHP treatment.
Another important remark, in my opinion, is the incorrect order of presentation of the results. First of all, the results of individual studies should be presented. Sections 2.1, 3.2, and 3.3 (there is also an error in numbering) should be placed before the summary of the significance of the individual factors examined (data in lines 96 - 111 and in Table 1).
The authors should also significantly improve the content of the conclusions.They refer only to a limited extent to the research results presented and contain too many general statements.
Other shortcomings include the lack of an explanation of the RBE abbreviation before its first use (it should be introduced on lines 89-90, similar to HPP).
The Authors should also correct the text fragment on lines 169-172, which contains repetition and a very general reference to other studies.
Reviewer 2 Report
Comments and Suggestions for Authors
The study presents promising results on chitosan-based active packaging enhanced with rice bran extract and HPP for extending the shelf life of Iberian ham. My comments are as follows:
- Please elaborate on the film formulation, particularly regarding crosslinking density and structural stability after high-pressure processing.
- FTIR or DSC data would support claims about polymer–plasticiser interactions and mechanical integrity.
- The instability of AP+HPP vs. AP alone warrants further rheological or SEM analysis to understand morphological changes.
- Migration tests for RBE should be included or referenced, taking into account food safety regulations.
- Consider evaluating release kinetics of active compounds under storage conditions to better link antimicrobial performance with polymer behaviour.
- Clarify whether the films can be integrated into existing packaging lines, as this would influence scalability.
Round 2
Reviewer 1 Report
Comments and Suggestions for Authors
The manuscript gels-3680771 significantly improved by the Authors, who rearranged its content and supplemented it with missing elements indicated in the review. The introduction contains reference to active packaging, which is also the object of the research presented in the article. The analysis of the properties of packaging films was supplemented with tests illustrating the changes in the material during the process of high hydrostatic pressure treatment. The conclusions presented in the article were based to a greater extent on the results of the conducted research. In my opinion the manuscript in its revised form may be considered for publication.
Reviewer 2 Report
Comments and Suggestions for Authors
Thank you for the thoughtful revision and for clearly acknowledging the study’s limitations. The manuscript is now significantly improved.